# Development and validation of deep learning algorithms for scoliosis screening using back images

Junlin Yang[1,10]*, Kai Zhang [2,3,10], Hengwei Fan[1,10], Zifang Huang[4,10], Yifan Xiang[2,10], Jingfan Yang[1,10], Lin He[3], Lei Zhang[3], Yahan Yang[2], Ruiyang Li[2], Yi Zhu[2,5], Chuan Chen [2,5], Fan Liu[3], Haoqing Yang[3], Yaolong Deng[1], Weiqing Tan[6], Nali Deng[6], Xuexiang Yu[7], Xiaoling Xuan[8], Xiaofeng Xie[8], Xiyang Liu[3] & Haotian Lin [2,9]*

Adolescent idiopathic scoliosis is the most common spinal disorder in adolescents with a prevalence of 0.5–5.2% worldwide. The traditional methods for scoliosis screening are easily accessible but require unnecessary referrals and radiography exposure due to their low positive predictive values. The application of deep learning algorithms has the potential to reduce unnecessary referrals and costs in scoliosis screening. Here, we developed and validated deep learning algorithms for automated scoliosis screening using unclothed back images. The accuracies of the algorithms were superior to those of human specialists in detecting scoliosis, detecting cases with a curve ≥20°, and severity grading for both binary classifications and the four-class classification. Our approach can be potentially applied in routine scoliosis screening and periodic follow-ups of pretreatment cases without radiation exposure.

[1] Spine Center, Xinhua Hospital Affiliated to Shanghai Jiaotong University School of Medicine, Shanghai, China. [2] State Key Laboratory of Ophthalmology, Zhongshan Ophthalmic Center, Sun Yat-sen University, Guangzhou, Guangdong, China. [3] School of Computer Science and Technology, Xidian University, Xi'an, Shanxi, China. [4] Department of Spine Surgery, the 1st Affiliated Hospital of Sun Yat-sen University, Guangzhou, Guangdong, China. [5] Department of Molecular and Cellular Pharmacology, University of Miami Miller School of Medicine, Miami, FL, USA. [6] Health Promotion Centre for Primary and Secondary Schools of Guangzhou Municipality, Guangzhou, Guangdong, China. [7] Department of Sports and Arts, Guangzhou Sport University, Guangzhou, Guangdong, China. [8] Xinmiao Scoliosis Prevention of Guangdong Province, Guangzhou, Guangdong, China. [9] Center for Precision Medicine, Sun Yat-sen University, Guangzhou, Guangdong, China. [10] These authors contributed equally: Junlin Yang, Kai Zhang, Hengwei Fan, Zifang Huang, Yifan Xiang, Jingfan Yang. *email: yjunlin@126.com; haot.lin@hotmail.com

Adolescent idiopathic scoliosis (AIS) is defined as a spinal curvature of 10° or more and of unknown etiology in persons aged 10 to 18 years. It is the most common spinal disorder in adolescents, with a prevalence of 0.5–5.2% worldwide[1–3]. Untreated scoliosis can progress before skeletal maturity, which influences body appearance, affects cardiopulmonary function, and even causes paralysis[4–6]. Consequently, school scoliosis screening (SSS) has been advocated for early detection of reversible spinal curves before their progression[7].

SSS involves different types of assessment methods, including appearance inspection, forward bending tests, scoliometer measurements, and individual Moiré topography, which are direct and easily accessible. However, the disadvantages of these methods are considerable and include their susceptibility to the subjectivity of screeners, time-consuming nature, and requirement for unnecessary radiography due to their low positive predictive values (PPVs)[8–11]. Although the US Preventive Services Task Force (USPSTF) changed the SSS recommendation from negative (D grade, discourage the use of screening programs) to neutral (I grade, uncertainty regarding the balance of the benefits and harm of the service) based on updated evidence in 2018, the deficiencies of traditional assessment methods in SSS remain unresolved[12–14].

Computer vision is recognized as a promising approach for image recognition in medicine due to its good performance in image information extraction[15–17]. Numerous tasks have been achieved by computer vision, including automated classification of eye diseases[18], identifying facial phenotypes of genetic disorders[19], and prediction of cardiovascular risk factors from fundus images[20]. As appearance features are evaluated by human screeners to detect scoliosis during SSS[8], these features may also be recognized by computer vision. Here, using unclothed back images, we developed deep learning algorithms (DLAs) and validated their feasibility and efficacy in the detection and severity grading of scoliosis. Large numbers of adolescents can be screened via telehealth examination and the exposure to radiation can be avoided.

## Results

**Algorithm training and internal validation**. The demographic information of the training and internal validation datasets is shown in Table 1. The entire block diagram and the architectures of Faster-RCNN and Resnet are shown in Fig. 1. The mean localization performance of Faster-RCNN was 100% (average interpolated precision, the standard deviation was 0). After the algorithms were established, five-fold cross-validation was applied to evaluate their performance. To detect cases with a curve ≥ 10°, algorithm 1 exhibited an average AUC of 0.946 (95% CI,

0.916–0.975), a sensitivity of 87.5% (95% CI, 81.2–93.8%), a specificity of 83.5% (95% CI, 77.6–89.4%), and a PPV of 86.2% (95% CI, 81.6–90.8%). For algorithm 2, the AUC, sensitivity, and specificity to detect cases with a curve ≥ 20° were 0.951 (95% CI, 0.933–0.970), 85.7% (95% CI, 83.4–88.1%), and 89.6% (95% CI, 86.2–93.0%), respectively, with a PPV of 89.1% (95% CI, 85.8–92.5%). The average accuracy of algorithm 3 was 80.0% (95% CI, 77.8–82.1%) for differentiating among the four groups. The internal validation results of the DLAs are shown in Fig. 2 and Supplementary Table 1.

**External Validation**. The diagnostic performance of the DLAs was further evaluated using the external validation dataset, and the demographic information of the external validation dataset is shown in Table 1. The professional screeners required ~30 min (range 19–40 min) to assess 400 back photos (4.5 s per photo), which was much longer than the time required for the DLAs (1.5 s per photo). The results showed significant differences between the DLAs and human expert panel in detecting scoliosis, detecting cases with a curve ≥20°, and curve severity grading; the P values were 0.022, <0.001, and <0.001, respectively. The AUCs of algorithms 1 and 2 were 0.811 (a sensitivity of 80.7%, a specificity of 58.0%) and 0.929 (a sensitivity of 84.0%, a specificity of 90.0%), respectively. The PPVs of these two algorithms were 85.2% and 89.4%, respectively. The DLAs exhibited higher accuracy than the human specialists in detecting scoliosis (algorithm 1, 75.0%; human, 72.4%) and identifying cases with a curve ≥ 20°, which require a brace or surgical treatment (algorithm 2, 87%; human, 81.9%). The accuracy of algorithm 3 was 55.5% for differentiating among the four groups, which was comparable to the highest accuracy (56.8%) of the four human experts and superior to the mean level (46.9%). The outcomes of the external validation are shown in Figs. 3 and 4 and Supplementary Table 2.

Heat maps suggested that the features contributing to intelligent discrimination of the DLAs were primarily in the scapular and lumbar regions (Fig. 5). The level of trunk asymmetry revealed in the heat maps was associated with the spinal curves of the patients. In addition, three binary classifications (20–44° and ≥45°; 25–44° and ≥45°; 0–44° and ≥45°) and another four classes of classifications (0–9°, 10–24°, 25–44°, ≥45°) were validated, as shown in Supplementary Fig. 1 and Supplementary Table 3. Moreover, we developed a cloud-based platform (Supplementary Fig. 2) embedded with the trained DLAs for self-screening in the Django web framework[21], which is available at http://www.spinecube.cn/login/en/. Users can access it freely.

## Discussion

This study showed that DLAs can be trained to detect scoliosis, identify cases with a curve ≥20°, and perform severity grading using unclothed back images with an accuracy, sensitivity, specificity, and PPV that are higher or comparable to those of human experts. Therefore, our algorithms can reduce the referral rate, costs, and time required for traditional scoliosis screening. Moreover, because the DLA method does not involve radiation exposure, it has the potential to be applied as a periodic follow-up tool for progression monitoring to avoid excessive X-ray exposure. To our knowledge, this is the first large and full-coverage (including healthy controls and various curvature severities) study on the intelligent detection of scoliosis. The efficiency of computer vision in the detection and classification of scoliosis was demonstrated using unclothed back images.

Several aspects may contribute to the superiority of our model. First, our DLAs achieve scoliosis screening using only unclothed

**Table 1 Demographic information of the training, internal validation and external validation datasets**

| Severity | Total number of subjects | Male | Female | Average severity | Average age (years old) |
|---|---|---|---|---|---|
| Training and internal validation datasets | | | | | |
| <10° | 745 | 306 | 439 | 6.3° | 15.2 |
| 10–19° | 938 | 312 | 626 | 15.2° | 13.9 |
| 20–44° | 780 | 197 | 583 | 26.9° | 15.4 |
| ≥45° | 777 | 214 | 563 | 56.2° | 16.1 |
| External validation dataset | | | | | |
| <10° | 100 | 43 | 57 | 6.3° | 14.6 |
| 10–19° | 100 | 32 | 68 | 16.1° | 14.1 |
| 20–44° | 100 | 41 | 59 | 28.4° | 15.3 |
| ≥45° | 100 | 45 | 55 | 51.3° | 15.7 |

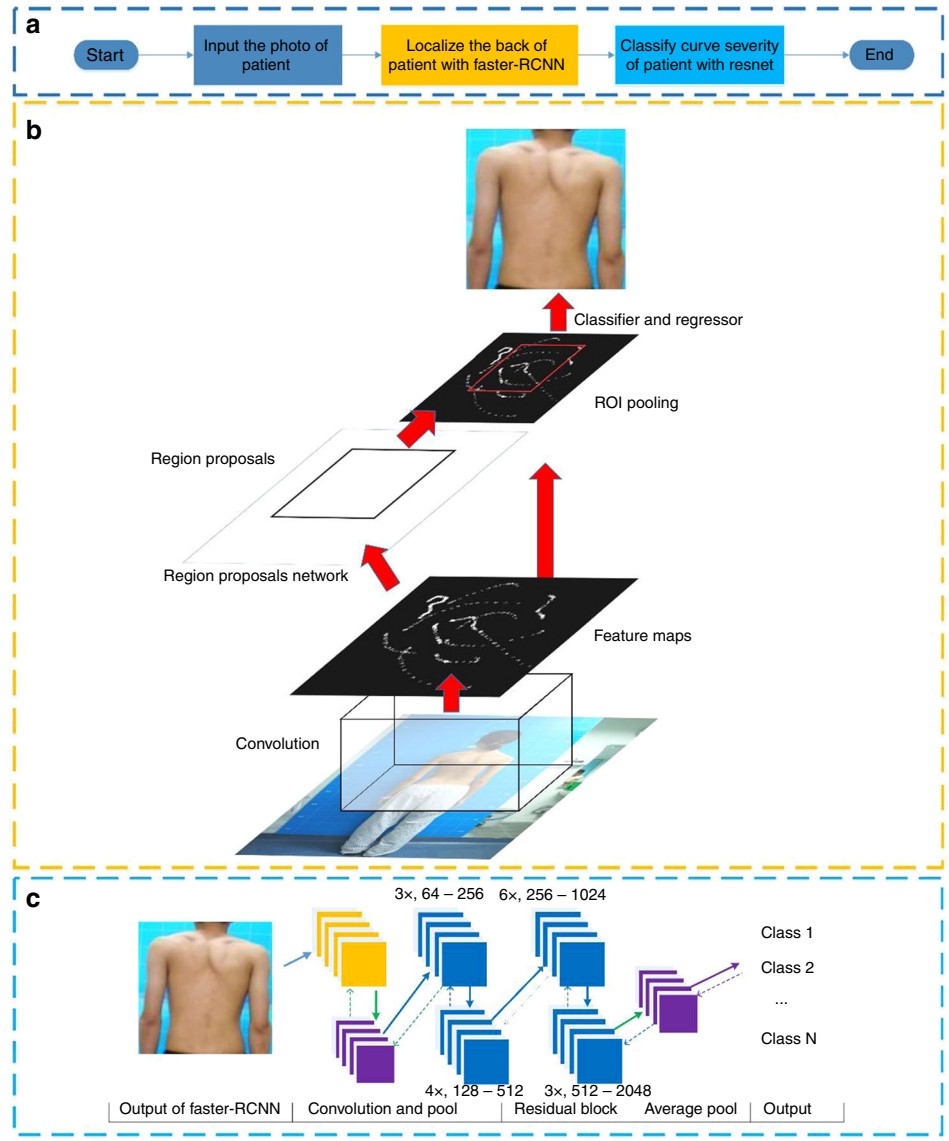

**Fig. 1** Details of the methods. **a** The entire DLA workflow; **b** The architecture and workflow of Faster-RCNN; **c** The architecture of Resnet, where the yellow squares in Restnet are the convolutional kernels in the convolutional layers and the purple squares in Resnet are pooling operations in the pooling layers

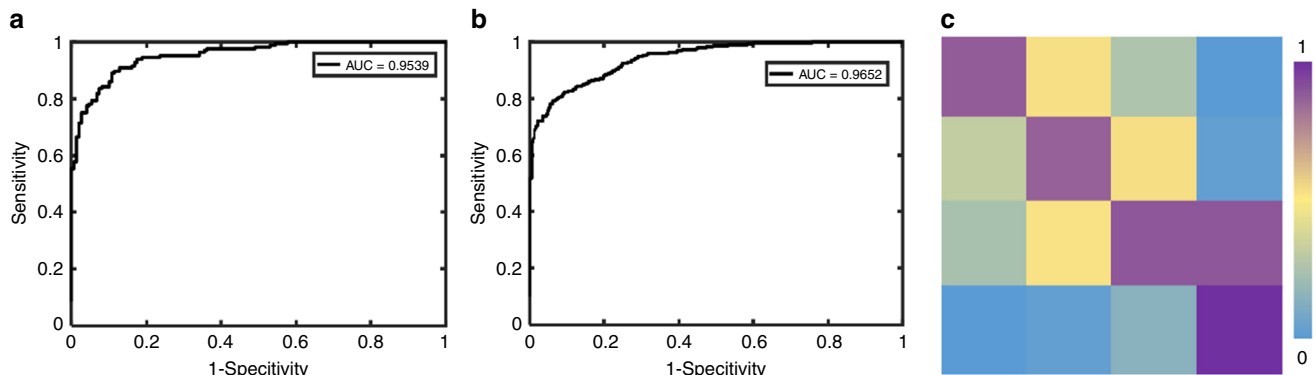

**Fig. 2** ROC curves and the accuracy, specificity, and sensitivity of the binary classification for the internal validation dataset. **a** The ROC curve and AUC of the DLAs for discerning whether the severity is ≥10°; **b** The ROC curve and AUC of the DLAs for discerning whether the severity is ≥20°; **c** The confusion matrix of the DLAs for the four classes of classifications (0–9°, 10–19°, 20–44°, ≥45°). The rows and columns represent the ground-truth label (<10°, 10–19°, 20–44°, ≥45° from top to bottom) and the predicted label (<10°, 10–19°, 20–44°, ≥45° from left to right)

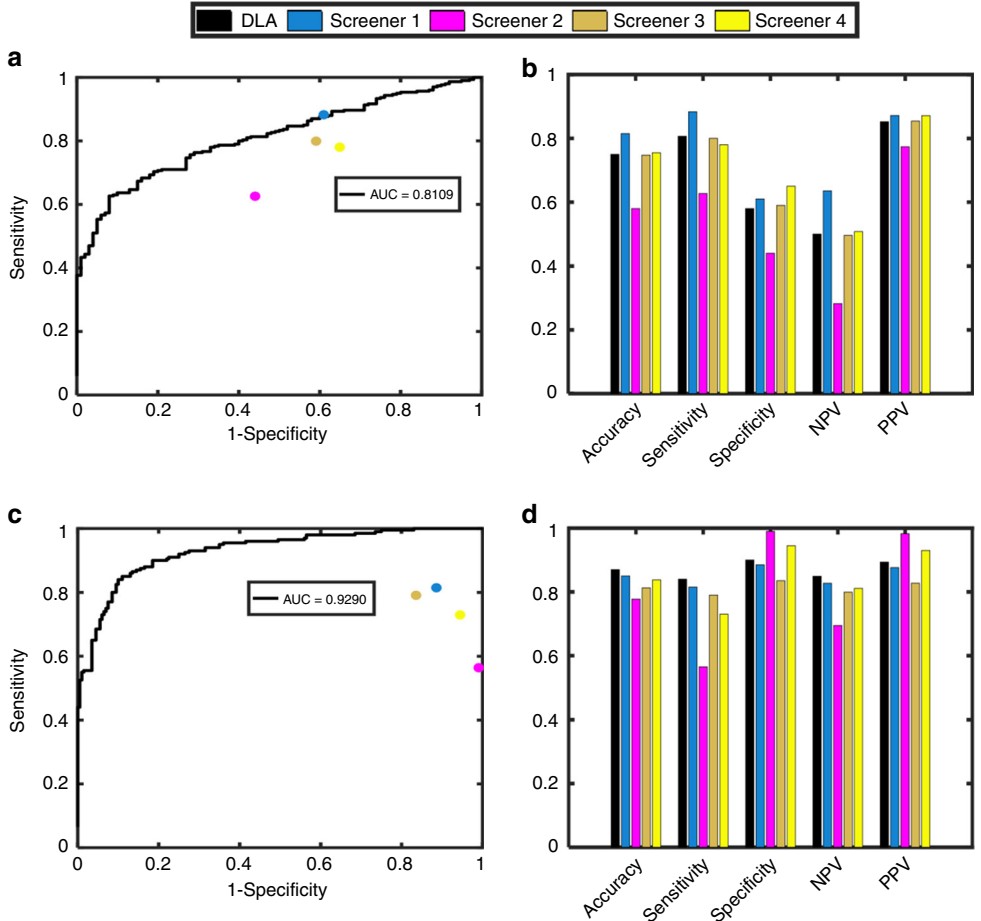

**Fig. 3** ROC curves and the accuracy, specificity, sensitivity, NPV, and PPV of the binary classification for the external validation dataset. **a** The ROC curve of the DLAs for discerning whether the severity is ≥10° and the performance of the human experts; **b** The accuracy, specificity, sensitivity, NPV, and PPV of the DLAs and human experts for discerning whether the severity is ≥10°; **c** The ROC curve of the DLAs for discerning whether the severity is ≥20° and the performance of the human experts; **d** The accuracy, specificity, sensitivity, NPV, and PPV of the DLAs and human experts for discerning whether the severity is ≥20°

back images and perform better than experts in terms of diagnostic accuracy. Second, our model has a clear criterion, so its resultant repeatability is higher than that of human experts, who grade scoliosis empirically based on superficial inspection. Third, our model can automatically process data and does not require human labor, so it performs more efficiently (DLAs vs human, 1.5 s vs 4.5 s per photo). We also note that one of our DLAs, algorithm 3, showed a moderate accuracy (55.5%) for distinguishing scoliosis severity, indicating the challenge of revealing subtle changes in scoliosis severity; however, even in these scenarios, our model still exhibits comparable performance to that of top human expert (56.8%). A previous study indicated that the performance of DLAs reached a plateau when the number of training cases increased to 60,000 (~17,000 referable images)[22]. Therefore, we believe that the performance of our algorithms will continue to improve as the number of training cases increases.

Various methods have been adopted in SSS programs to reduce the referral rate[8]. In the study by Karachalios et al.[23], four single-modality screening methods were analyzed in the same cohort. The sensitivity of these four methods ranged from 84.4% to 100%; the specificity ranged from 78.1% to 93.4%; but the PPVs were low (ranging from 4.8% to 13.3%). The PPV was improved when two or more screening methods were combined as reported by Luk et al. (forward bending test with a scoliometer and Moiré screening, PPV = 81%) and Yawn et al. (forward bending test with a scoliometer, PPV = 29.3%)[24,25]. However, more screening

devices and time are required when two or more screening methods are adopted in combinations, especially when screening a large population. To detect cases with curves of 20° or greater, Luk et al. reported a PPV of 39.8%, a sensitivity of 91.0%, and a specificity of 97.5%, and Yawn et al. reported a PPV of 17.0%, a sensitivity of 64.0% and a specificity of 67.2%[24,25]. Most of these screening methods, whether used alone or in combination, had lower PPVs for detecting scoliosis cases (curves ≥ 10°) and cases with a curve ≥ 20° than the DLAs tested in the present study. Our present study using DLAs achieved PPVs of 85.2% (a sensitivity of 80.7% and a specificity of 58.0%) and 89.4% (a sensitivity of 84.0% and a specificity of 90.0%) when identifying scoliosis and cases with a curve ≥ 20°, which are far better than those reported previously and are comparable to those achieved by specialists for algorithm 1 (mean PPV 84.3%, range 77.4–87.2%) and algorithm 2 (mean PPV 90.4%, range 82.7–98.3%) in the external validation. Therefore, these algorithms can reduce the referral rate and the number of cases unnecessarily exposed to radiation due to false-positive results.

The main drawback of using the Cobb angle via radiography is the increased risk of cancer, which impedes its routine application in scoliosis screening[26–30]. Several machine learning methods have been designed to detect spinal deformity using the trunk surface defined by various techniques, including optical digitizing systems[31–33], orthogonal maps[34], the surface topography technique[35], laser scanners[36,37], and the Quantec system[38]. However,

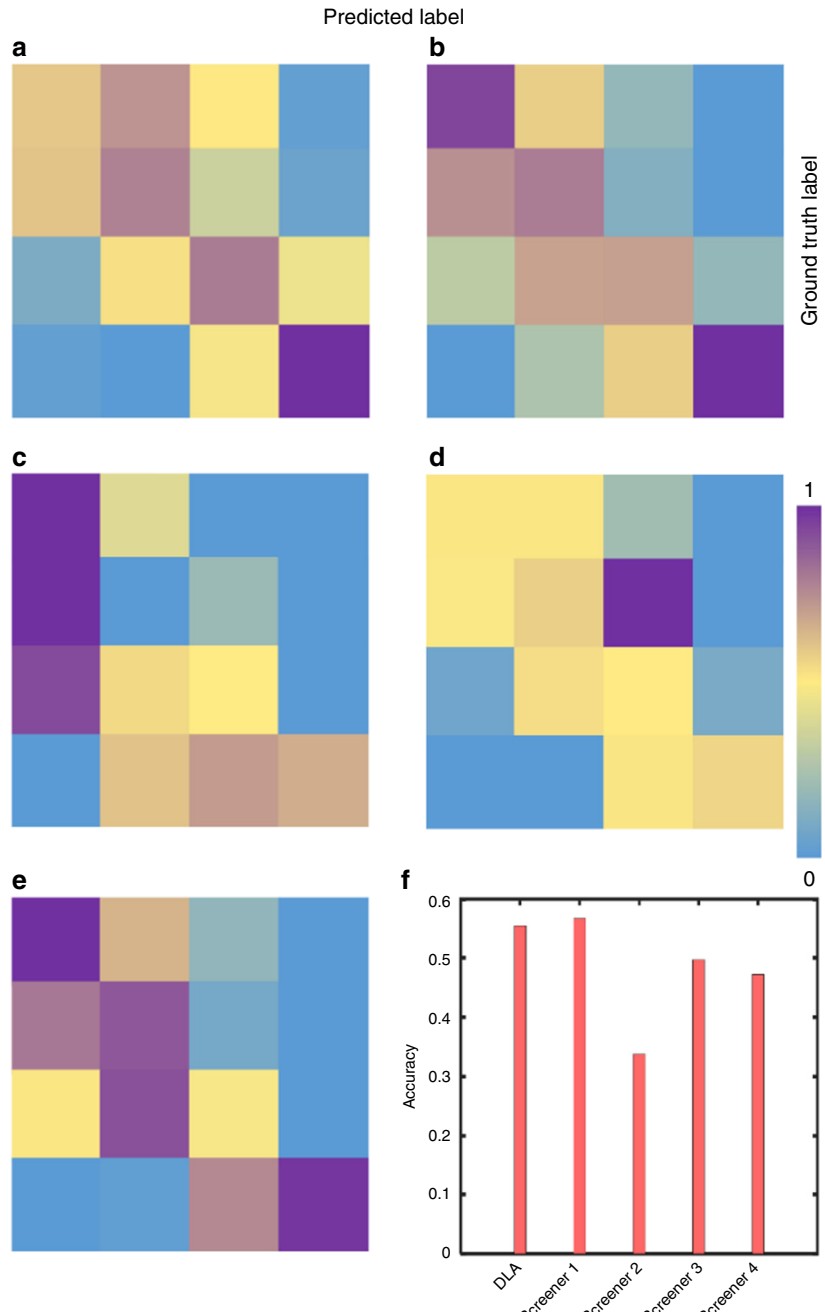

**Fig. 4** The confusion matrix of the four classes of classification and their accuracy in the external validation dataset. The rows and columns represent the ground-truth label (<10°, 10–19°, 20–44°, ≥45° from top to bottom) and the predicted label (<10°, 10–19°, 20–44°, ≥45° from left to right). **a** The confusion matrix of the DLAs for classifying the severity (four classes of classifications: <10°, 10–19°, 20–44°, ≥45°); **b**–**e** The confusion matrix of the human experts for classifying the severity (four classes of classifications: <10°, 10–19°, 20–44°, ≥45°); **f** The accuracy of the DLAs and human experts in classifying the severity (four classes of classifications: <10°, 10–19°, 20–44°, ≥45°)

these methods still cannot be widely applied due to small scoliosis datasets and the lack of healthy controls, the high demand for specialized equipment, and the time-consuming nature of these methods. In contrast, our DLAs require only unclothed back photos for diagnosis and do not require radiation exposure.

Another argument against SSS is the requirement for manpower, equipment, and financial resources. The screening costs include the expenditure for training screeners, salaries of screeners, and purchasing screening devices[39]. The screening cost is from USD 1.81 to USD 13.61 per student in different SSS programs[40,41]. The cost increased when students were followed

up before their skeletal maturity. A Rochester study showed that the cost increased to USD 34.40 per student when follow-up was included[39]. Luk et al. reported that the average cost of screening one student for the whole adolescent period was USD 17.94, which increased to USD 20.02 and USD 54.63 per student when diagnostic and medical care costs were included[10]. Our web-based self-screening approach will greatly reduce screening costs, as only network maintenance or a simple screening device with DLAs is needed.

Clinical signs of unevenness on back images, including shoulder height, scapular prominence, and truncal shift, can serve

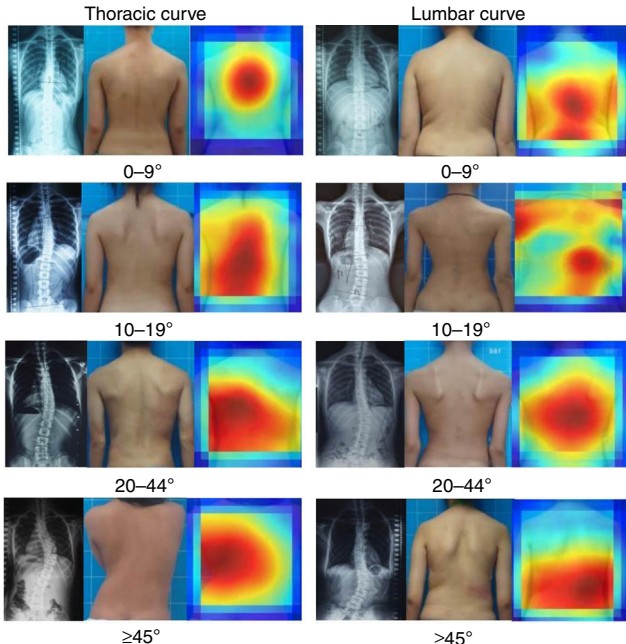

**Fig. 5** Heat maps illustrating which parts of the body contributed to the classification results. Heat maps suggested that the features contributing to intelligent discrimination by the DLAs were primarily in the scapular and lumbar regions. The level of trunk asymmetry revealed in the heat maps was associated with the spinal curves of the patients

as important clues indicating the need for referral of a scoliosis patient to radiography[24]. Our heat map (Fig. 5) showed that the scapular and lumbar regions are critical for the predictive accuracy of the DLAs, providing a theoretical basis for screening and classifying scoliosis according to body surface characteristics. This predictive method is similar to human screening practice and has the capacity to distinguish scoliosis severity between two or more categories. Moreover, further study of surface parameters recognized by DLAs may help improve current manual screening methods.

This study has several limitations. First, all scoliosis cases in the training set were verified by experts to ensure the diagnosis of idiopathic scoliosis, and only those who underwent corrective surgery received detailed examinations (whole-spine MRI, CT, or electrophysiological examination). Therefore, the cases screened out by the DLAs may have mixed etiologies, such as congenital scoliosis, Marfan syndrome, or neuromuscular scoliosis. Second, whole-spine X-rays of a healthy population (Cobb angle < 10°) were substituted by full-spine ultrasound images to avoid radiation exposure because a large dataset of X-ray images from a healthy population was unavailable. Third, the accuracy of the algorithm needs to be improved with more training data or multidimensional photos, and the application of the DLA platform requires further validation in multicenter and multiethnic trials.

## Methods

**Data collection**. The training dataset included images from 3240 patients (including 1029 male and 2211 female) with labeled back images and whole-spine standing posterior-anterior X-ray images or ultrasound images from the databases of three institutions (including ~1.2 million screened school students, with 6500 conservatively treated and 2000 surgically treated patients with scoliosis). Of these images, 2495 were obtained from subjects with scoliosis and 745 were obtained from normal controls. All the subjects were Chinese with an age range from 10 to 20 years old, subjects with nontrue scoliosis (such as scoliosis caused by pain or by leg discrepancy, etc.), other spine diseases or some other back abnormalities (such as soft tissue mass, thoracic cage diseases, etc) were excluded. The datasets used for

external validation consisted of data from 400 individuals, including 100 normal cases and 300 scoliosis cases.

This study was approved by the Ethics Committee of Xinhua Hospital, Xinmiao Spine Clinic, and the 1st Affiliated Hospital of Sun Yat-sen University. The test between the DLAs and the orthopedists was registered with ClinicalTrials.gov (identifier: NCT03773458). The back images for algorithm training and validation were collected from patients screened or treated for scoliosis at these three institutions. All back images were obtained with the subjects standing naturally and disrobing above the hip. Various cameras were used for image acquisition. The requirement for informed consent was waived because of the retrospective nature of the fully anonymized images. Our study was approved by the institutional review board (IRB: XHEC-KJB-2018–024) and was conducted in accordance with the Declaration of Helsinki.

**Annotation Process**. Summaries of each dataset are shown in Supplementary Table 4. Each subject had a definitive diagnosis based on a whole spine standing posterior-anterior radiograph or a full-spine ultrasound image obtained with a Scolioscan system (Model SCN801; Telefield Medical Imaging Ltd, Hong Kong). Normal cases in this study were recruited volunteers who were diagnosed by ultrasound images (VPI-SP method) in radiation-free evaluations, which showed a low mean difference ($d = -1.9°$) compared with Cobb's method for X-ray film in mild to moderate scoliosis.[41] Cobb angles of the spine in all datasets were measured by two separate spine surgeons with 10 years' experience based on medical images, and a senior spine specialist, who has been treating AIS for more than 25 years, was consulted if unanimous agreement was not achieved. Meanwhile, all back images were labeled with tool (https://github.com/tzutalin/labelImg). The intraobserver and interobserver correlation coefficients for the measurement of Cobb angles were 0.974 and 0.917, respectively. The subjects were divided into different groups according to curve severity: Group 1, Cobb angle < 10°; Group 2, 10° ≤ Cobb angle < 20°; Group 3, 20° ≤ Cobb angle < 45°; and Group 4, Cobb angle ≥ 45°. Group 1 represented nonscoliotic subjects, and the other groups represented patients with scoliosis. Patients in groups 2, 3, and 4 were candidates with different management strategies (Group 2, periodic observation or physiotherapy; Group 3, bracing; and Group 4, corrective surgery).

**Architecture of the DLAs**. The regions of interest (from neck to hip) were first automatically localized using Faster-RCNN[42]. Resnet with 101 layers[43] was developed to recognize the characteristics of each group[43]. The preprocessed images were then input into the Resnet to extract high-level features for binary classifications and multiclass classifications based on the groups mentioned above. The diagrammatic sketches of Faster-RCNN and Resnet with 101 layers are shown in Fig. 1c and Supplementary Fig. 3a. Moreover, the architectures of the residual block and batch normalization block are shown in Supplementary Fig. 3b, c. Deep learning is a type of neural network that can imitate the information summarizing ability of the human brain via multilayer connections among neurons. The DLAs were trained to differentiate the unseen images. Algorithm 1 was trained to distinguish scoliotic (groups 2, 3, and 4) from nonscoliotic (group 1) subjects; algorithm 2 was used to identify potential subjects in need of a brace or surgical treatment (groups 3 and 4); and algorithm 3 was designed to divide the multiclass classifications of sorting images into 4 groups (0–9°; 10°–19°; 20°–44°; ≥45°). The examples (unclothed back, X-ray and ultrasound images) of these four severities are provided in Supplementary Fig. 4.

**Statistics and reproducibility**. The algorithms were assessed independently for each purpose. The $k$-fold cross validation ($k = 5$) technique was used to evaluate the capacity of the trained DLAs to correctly classify the back images. The external validation dataset was used to further determine the performance of the neural network. The performances in image classification were compared between the DLAs and four professional screeners with an external validation dataset, which was registered with ClinicalTrials.gov (identifier: NCT03773458). The detailed information of these four screeners is shown below:

**Professor Junlin Yang (senior spine specialist)**. Professor Yang is a famous orthopediac surgeon in China, and has engaged in clinical and basic orthopedic research for ~30 years (with spinal deformity experience ~20 years). He has performed >2000 scoliosis surgeries, >5000 conservative scoliosis treatment and more than 20,000 of scoliosis outpatient consultations.

**Doctor Zifang Huang and Jingfan Yang (two junior spine surgeons)**. Doctors Huang and Yang are both spine surgeons with more than ten-year experience in screening, conservative treatment and surgery in spinal deformity. They have participated in >500 scoliosis surgeries, >1000 conservative scoliosis treatments and >3000 of scoliosis outpatient consultations.

**Xiaoling Xuan**. As a trained nurse screener, Xiaoling Xuan has engaged in more than three years of school scoliosis screening in Guangdong Province, and screened for scoliosis in >10,000 adolescent students.

For 5-fold cross validation and external validation, the sensitivity, specificity, accuracy, and area under the receiver operating characteristic (AUC) curve of the DLAs were calculated. The number of true-positive (TP), true-negative (TN), false-positive (FP), and false-negative (FN) cases were counted, and the corresponding PPV and negative predictive value (NPV) were calculated[44,45]. The values can be obtained as follows:

$$\text{Accuracy} = (TP + TN)/(P + N)$$

$$\text{Sensitivity(TPR(true positive rate))} = TP/(TP + FN) = TP/P$$

$$\text{Specificity} = TN/(TN + FP) = TN/N$$

$$\text{NPV(negative predictive value)} = TN/(TN + FN)$$

$$\text{PPV(positive predictive value)} = TP/(TP + FP)$$

where, N and P are the numbers of negative samples and positive samples, respectively. The threshold for calculating the accuracy, sensitivity, specificity, NPV and PPV was 0.5. In binary classification problem, if the largest one out of two classification probabilities for a sample is larger than 0.5, then the predicted label for this sample was the class corresponding to this largest probability. The performances of the DLAs in the external validation were compared with those of professional screeners using descriptive statistics. All of the statistical tests in our study were 2-sided, and a P-value less than 0.05 was considered significant. All analyses were performed with MATLAB (Version R2016a, MathWorks, http://www.mathworks.com).

The performances of all four screeners and DLAs were independently evaluated with 400 samples (unclothed back images) in three different scenes using the above mentioned performance measurements. Screeners were told that the options included <10° and ≥10° in the first scene. Similarly, Screeners were told that the options included <20° and ≥20° in the second scene and that options included <10°, 10–19°, 20–44° and ≥45° in the third scene.

**Contribution analysis and website building**. We also analyzed the contributions of body parts to the discriminative capacity of the DLAs using class activation mapping (CAM)[46]. Regions of interest are shown as colored areas on heat maps. Additionally, a self-screening website (https://www.spinecube.cn) was built for free access. A user can upload his/her back image according to the instructions on the website, and the output report includes a screening result and a referral recommendation (radiography or online follow-up).

All the methods mentioned above were carried out with the Berkeley Vision and Learning Center (BVLC) deep-learning framework (Caffe[47]) on a computer with four TITAN Xp graphical processing units. For Faster-RCNN, the localization performance was evaluated with average precision (AP[48,49]) and four-fold cross-validation. For Resnet, the classification performance was evaluated by assessing the accuracy, sensitivity, specificity, receiver operating characteristic (ROC) curve, and AUC curve, as well as by five-fold cross-validation. All DLAs were fine-tuned from pretrained models, specifically, the hyperparameters of all DLAs were initialed from an existing model that were trained with IMAGENET dataset[50]. We trained Alexet, VGG16, Inception-V4 and Resnet-101 with the same dataset, and we chose Resnet-101 to complete this task.

**Reporting summary**. Further information on research design is available in the Nature Research Reporting Summary linked to this article.

## Data availability

The training data that support the findings of this study contain sensitive patient information and, due to legal regulations and institutional ethics policies enacted, can no longer be made publicly available. However, to support scientific collaboration, data may be accessible to qualified researchers under a formal collaborative agreement. All requests for access must be submitted to the corresponding author, Junlin Yang, and are subject to approval by our institution's ethics committee.

## Code availability

The configuration files and instructions required to implement the methods described in this study are available in the Spinecube repository on GitHub at https://github.com/Hugo0512/Spinecube. The implementation relies on two key modules executed in the Caffe framework: the Faster-RCNN module, based on the work of Girshick (https://github.com/rbgirshick/py-faster-rcnn), and the ResNet-101 model architecture, adapted from He et al. (https://github.com/KaimingHe/deep-residual-networks). The repository provides specific guidance on environment setup and the parameters used for these frameworks.

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

## Acknowledgements

Gratitude is due to the National Super Computer Center in Guangzhou (NSCCGZ), Sun Yat-sen University and Tianhe-2. This study was funded by the National Key R&D Program of China (2018YFC0116500), the National Natural Science Foundation of China (81822010), the Science and Technology Planning Projects of Guangdong Province (2018B010109008), the National Natural Science Foundation Fund for Overseas and HongKong and Macao Scholars Joint Research Project (81828007), the Key Project of Transformational Medicine Cross-Research Fund of Shanghai Jiaotong University (ZH2018ZDB04), the Guangdong Science and Technology Planning Project (2014B020212021 and 2015A070710007), and the Key Research and Development Program of Guangdong Province (2018B010109008). The funders of the study had no role in study design, data collection, data analysis, data interpretation, or writing of the report. The corresponding author had full access to all the data in the study and had final responsibility for the decision to submit for publication.

## Author contributions

H.T.L. and J.L.Y. designed the study; H.W.F., J.F.Y., Y.L.D., W.Q.T., N.L.D., J.L.Y., Y.H.Y., R.Y.L., and X.X.Y. collected the data and annotated the data; Z.F.H., X.L.X., and X.F.X. performed the statistical analysis; K.Z., L.H., F. L., L.Z., H.Q.Y., and X.Y.L. were responsible for coding and analysis and for completing the experimental results. Y.F.X., Z.F.H., K.Z. and H.W.F. drafted the manuscript; and H.T.L., Y.F.X., Y.Z., and C.C. critical revised the manuscript for important intellectual content; All authors discussed the results and commented on the manuscript.

## Competing interests

The authors declare no competing interests.
