## [Peer Review File · Communications Biology]

Reviewers' comments:

Reviewer #1 (Remarks to the Author):

What are the major claims of the paper?

In this paper authors claim to demonstrate how computer vision algorithms can be used to screen for scoliosis using unclothed back images.

Are they novel and will they be of interest to others in the community and the wider field? Authors demonstrated that this work is novel in its coverage and ability to screen for specific conditions, as well as they presented a way to deploy this work for real usage. This type of work may be valuable to others in the community as they may learn that AI tools are widely available for the use in different fields and may demonstrate additional applications to other medical conditions as well as open the door for process automation using AI.

If the conclusions are not original, it would be helpful if you could provide relevant references.

The conclusions seem to be original.

Is the work convincing, and if not, what further evidence would be required to strengthen the conclusions?

There are some missing parts that may help better understand the results. Authors report AUC, sensitivity, specificity and accuracy with mixed usages, it is not clear how each was obtained. For example what threshold was used to calculate the accuracy? It will also be helpful to see the ROC curve and have some handle of the potential bias due to having one class with 100 individuals vs 300 (algo 1). In addition, there seems like it will be useful to extend the explanation on the comparison to human experts. For example what is the level of expertise, how one result was obtained (is it a majority voting?).

On a more subjective note, do you feel that the paper will influence thinking in the field?

Yes, I think that by demonstrating novel applications of AI for health care will influence more research on using such tools and may increase the overall impact.

Please feel free to raise any further questions and concerns about the paper.

We would also be grateful if you could comment on the appropriateness and validity of any statistical analysis, as well as the ability of a researcher to reproduce the work, given the level of detail provided.

It seems like the work can be reproduced by having access to the training data.

I think statistical analysis should be better. It can be more aligned across experiments, with relevant adjustments. More explanation is needed on the nature of all populations, age, gender, ethnic background, etc. This should be done for all groups, where it is much valuable to know the distribution and to make sure there are no significant differences between the groups and the network indeed learned discriminative features.

Reviewer #2 (Remarks to the Author):

Summary of the article: This study focused on clinical validation of various stages of scoliosis using deep learning approaches.

Comments:

1. Even though the literature review, data collection, and contributions are reported in this article. But, I think the presentation of the article should be revised. Related works, Data collection, and Contributions should be in separate sections.
2. Related work should be extended with mentioning the limitation of the existing related studies. This can make the contributions stronger.
3. Please explain the motivation and mention the main contribution and three sub-contributions of this study in separate section.
4. Data annotation process should be explained clearly and in details since the study aimed to validate different stages of scoliosis.
5. Even though this paper has clinical contribution, but it is important to explain the deep learning model briefly. I suggest to use the fact that Neural Networks inspired by human brain.
6. The network architecture and fine-tuning process should be explained briefly.
7. The reason why these architectures are chosen should be mentioned.

Our point-by-point responses are as follows:

Reviewer #2: *What are the major claims of the paper?*

in this paper authors claim to demonstrate how computer vision algorithms can be used to screen for scoliosis using unclothed back images.

Are they novel and will they be of interest to others in the community and the wider field?

Authors demonstrated that this work is novel in its coverage and ability to screen for specific condition, as well as they presented a way to deploy this work for real usage. This type of work may be valuable to others in the community as they may learn that AI tools are widely available for the use in different fields and may demonstrate additional applications to other medical conditions as well as open the door for process automation using AI.

If the conclusions are not original, it would be helpful if you could provide relevant references.

The conclusions seem to be original.

Thank you very much for your recognition! We sincerely appreciate your scrupulous and constructive suggestions, which were valuable for improving the quality of our article.

Comment (1): *Is the work convincing, and if not, what further evidence would be required to strengthen the conclusions?*

There are some missing parts that may help better understand the results. Authors report AUC, sensitivity, specificity and accuracy with mixed usages, it is not clear how each was obtained. For example what threshold was used to calculate the accuracy? It will also be helpful to see the ROC curve and have some handle of the potential bias due to having one class with 100 individuals vs 300 (algo 1). In addition, there seems like it will be useful to extend the explanation on the comparison to human experts. For example what is the level of expertise, how one results was obtained (is it a majority voting?).

Response: Thank you very much for your advices.

First, we supplemented the formulas for obtaining the accuracy, sensitivity, specificity, NPV and PPV. In current research, the threshold for calculating the Accuracy, Sensitivity, Specificity, NPV and PPV was 0.5. In binary classification problem, if the largest one out of two classification probabilities for a sample was larger than 0.5, then the predicted label of this sample is the class corresponding to this largest probability (Page 14 Lines 277 - 287).

Second, the ROC curves obtained in all binary classification problems (internal and external validation) were shown in Fig. 2 and Fig. 3 (Page 22 Line 471, Page 23 Line 485).

Third, In the external validation dataset, there are a total of 400 samples (100 samples are smaller than 10° , 100 samples are larger than 10° and smaller than 20° , 100 samples are larger than 19° and smaller than 45° , and 100 samples are larger than or equal to 45°). We repeatedly used these 400 samples to test the the DLA 1, DLA 2 and DLA 3.

Fourth, we ask four screeners to evaluate the severity of the 400 samples that were used to test the performance of the DLAs in three different scenes. Screeners were told that the options included ' $<10^\circ$ ' and ' $\geq 10^\circ$ ' in the first scene, ' $< 20^\circ$ ' and ' $\geq 20^\circ$ ' in the second scene and ' $< 10^\circ$ ', ' $10 - 19^\circ$ ', ' $20 - 44^\circ$ ' and ' $\geq 45^\circ$ ' in the third scene. Then we obtained the performance indicators for the four screeners and compared the accuracy, sensitivity, specificity, NPV and PPV of screeners and DLAs. All four screeners and DLAs were independently evaluated with the 400 samples (unclothed back images) in three different scenes using above mentioned performance indicators. The level of expertise of the four screeners has been included in the revised manuscript, the senior spine specialist is Prof. Junlin Yang, the junior spine surgeons are Dr. Zifang Huang and Dr. Jingfan Yang, and the trained nurse screener is Xiaoling Xuan. Their detailed information is shown below:

Professor Junlin Yang (senior spine specialist): he is a famous orthopediac surgeon in China, Professor Yang has engaged in clinical and basic orthopedic research for approximately 30 years (with spinal deformity experience approximately 20 years). He has performed >2000 scoliosis surgeries, >5000 conservative scoliosis treatment and more than 20,000 of scoliosis outpatient consultations.

Doctor Zifang Huang and Jingfang Yang (two junior spine surgeons): Doctors Huang and Yang are both spine surgeons with more than ten years experience in screening, conservative treatment and surgery in spinal deformity. They have participated in >500 scoliosis surgeries, >1000 conservative scoliosis treatments and more than 3,000 of scoliosis outpatient consultations.

Xiaoling Xuan: As a trained nurse screener, Xiaoling Xuan has engaged in more than three years of school scoliosis screening in Guangdong Province, and screened for scoliosis in more than 10,000 adolescent students (Page 15 Line 291 – 296, Page 13 Line 258 - 272).

Comment (2): On a more subjective note, do you feel that the paper will influence thinking in the field?

Yes, I think that by demonstrating novel applications of AI for health care will influence more research on using such tools and may increase the overall impact.

Response: Thank you for your recognition.

Comment (3): Please feel free to raise any further questions and concerns about the paper.

We would also be grateful if you could comment on the appropriateness and validity of any statistical analysis, as well the ability of a researcher to reproduce the work, given the level of detail provided.

It seems like the work can be reproduced by having access to the training data.

I think statistical analysis should be better. It can be more aligned across experiments, with relevant adjustment. More explanation is needed on the nature of all populations, age gender ethnic background etc. This should be done for all groups, where is much valuable to know the distribution and to make sure there are no significant difference between the groups and the network indeed learned discriminative features.

Response: This research can be easily reproduced with the open-source tool and project in Github. We have provided a “Data availability” section to give readers a chance to request for training data.

The demographic information of the datasets used to train the DLAs has been included in the revised manuscript. All of subjects were Chinese people, aged from 10 to 18 years old, including 1,029 males and 2,211 females from the school scoliosis screening and conservative database (Xinmiao Scoliosis Prevention of Guangdong Province) and AIS surgery correction database (the 1st Affiliated Hospital of Sun Yat-sen University and Xinhua Hospital Affiliated to Shanghai Jiaotong University School of Medicine), and this information is now included in the RESULTS section of the article. Moreover, we think the nature of the external validation dataset will not impact the result, because the DLAs and human experts in the external validation used the same dataset, which was also from the same database as the internal validation dataset (Page 5 Line 77, 92; Page 27 Line 531).

Reviewer #2: Summary

Summary of the article: This study focused on clinical validation of various stages of scoliosis using deep learning approaches.

Thank you very much for your recognition! We sincerely appreciate your scrupulous and constructive suggestions, which were really valuable for improving the quality of our articles. We asked an English-language editing company to improve the language of our manuscript. Moreover, we also asked an English teacher to improve the language of our manuscript.

Comment (1): Even though the literature review, data collection, and contributions are reported in this article. But, I think the presentation of the article should be revised. Related works, Data collection, and Contributions should be in separate sections.

Response: Thank you for your suggestion. We have rearranged our manuscript and supplemented some subheadings in the Discussion section to guide readers and facilitates the understanding of the contribution made in this paper. Moreover, the comparison between our work and previous works can also be easily interpreted. Finally, the data collection section was moved to METHODS section (Page 11 Line 200, Page 12 Line 218; Page 13 Line 252).

Comment (2): Related work should be extended with mentioning the limitation of the existing related studies. This can make the contributions stronger.

Response: Thank you for your constructive advices, we have added relevant content to explain to readers the limitations of previous studies and to strengthen the explanation of the contributions made by present study (Page 8 Lines 150 - 152; Page 9 Lines 165 -169).

Comment (3): Please explain the motivation and mention the main contribution and three sub-contributions of this study in separate section.

Response: Thank you for your constructive advice, we have added a description of the motivation of this research in manuscript and four subheadings were added in the Discussion section to guide readers to quickly grasp the main contributions of this research (Page 5 Line 73; Page 7 Line 125; Page 8 Line 139; Page 9 Line 160).

Comment (4): Data annotation process should be explained clearly and in details since the study aimed to validate different stages of scoliosis.

Response: Thank you for your constructive advices. The annotation process was depicted and supplemented in the METHODS section (Page 12 Lines 225 - 228).

Comment (5): Even though this paper has clinical contribution, but it is important to explain the deep learning model briefly. I suggest to use the fact that Neural Networks inspired by human brain.

Response: Thank you for your constructive advices. We briefly depicted the mechanism of deep neural networks to help readers understand its advantages (Page 13 Lines 243 - 245).

Comment (6): The network architecture and fine-tuning process should be explained briefly.

Response: Thank you for your advice. Diagrammatic sketches of Faster-RCNN and Resnet with 101 layers have been supplemented and The fine-tune process has also been explained (Page 16 Lines 309 - 3011).

Comment (7): The reason why these architectures are chosen should be mentioned.

Response: Thank you for your constructive advice. We tested Alexet, VGG16, Inception-V4 and Resnet-101 with same dataset and the Resnet-101 performed best. We have supplemented the reason why we choose Resnet with 101 layers (Page 16 Lines 311 - 313).

Finally, thank you again for giving us this opportunity to revise our manuscript. We are very grateful for your great suggestions and help. We truly cannot thank you enough for shepherding our paper toward publication.

REVIEWERS' COMMENTS:

Reviewer #1 (Remarks to the Author):

Thank you for addressing my comments, good luck with this publication.